# PPAR Gamma: From Definition to Molecular Targets and Therapy of Lung Diseases

**DOI:** 10.3390/ijms22020805

**Published:** 2021-01-15

**Authors:** Márcia V. de Carvalho, Cassiano F. Gonçalves-de-Albuquerque, Adriana R. Silva

**Affiliations:** 1Laboratório de Imunofarmacologia, Instituto Oswaldo Cruz, Fundação Oswaldo Cruz (FIOCRUZ), Rio de Janeiro 21040-900, Brazil; marciavcarvalho@gmail.com; 2Programa de Pós-Graduação em Biologia Celular e Molecular, Instituto Oswaldo Cruz, Fundação Oswaldo Cruz (FIOCRUZ), Rio de Janeiro 21040-900, Brazil; 3Laboratório de Imunofarmacologia, Universidade Federal do Estado do Rio de Janeiro (UNIRIO), Rio de Janeiro 20211-010, Brazil; 4Programa de Pós-Graduação em Biologia Molecular e Celular, Universidade Federal do Estado do Rio de Janeiro (UNIRIO), Rio de Janeiro 20211-010, Brazil

**Keywords:** lung injury, acute respiratory distress, inflammation, PPARγ, molecular targets

## Abstract

Peroxisome proliferator-activated receptors (PPARs) are members of the nuclear receptor superfamily that regulate the expression of genes related to lipid and glucose metabolism and inflammation. There are three members: PPARα, PPARβ or PPARγ. PPARγ have several ligands. The natural agonists are omega 9, curcumin, eicosanoids and others. Among the synthetic ligands, we highlight the thiazolidinediones, clinically used as an antidiabetic. Many of these studies involve natural or synthetic products in different pathologies. The mechanisms that regulate PPARγ involve post-translational modifications, such as phosphorylation, sumoylation and ubiquitination, among others. It is known that anti-inflammatory mechanisms involve the inhibition of other transcription factors, such as nuclear factor kB(NFκB), signal transducer and activator of transcription (STAT) or activator protein 1 (AP-1), or intracellular signaling proteins such as mitogen-activated protein (MAP) kinases. PPARγ transrepresses other transcription factors and consequently inhibits gene expression of inflammatory mediators, known as biomarkers for morbidity and mortality, leading to control of the exacerbated inflammation that occurs, for instance, in lung injury/acute respiratory distress. Many studies have shown the therapeutic potentials of PPARγ on pulmonary diseases. Herein, we describe activities of the PPARγ as a modulator of inflammation, focusing on lung injury and including definition and mechanisms of regulation, biological effects and molecular targets, and its role in lung diseases caused by inflammatory stimuli, bacteria and virus, and molecular-based therapy.

## 1. Introduction

Peroxisome proliferator-activated receptors (PPARs) are ligand-dependent transcription factors of the nuclear receptor superfamily that regulate the expression of specific target genes involved in energy and lipid metabolism, adipogenesis and inflammation. In mammals, the PPAR subfamily consists of three members: PPAR-α, PPAR-β/δ and PPAR-γ [1,2]. PPARα (also called NR1C1) was initially identified as an orphan receptor activated by various peroxisome proliferators [3]. PPARβ/δ (NR1C2) and PPARγ (NR1C3) were subsequently cloned as related receptors that are activated by distinct peroxisome proliferators [4].

PPARγ has two major isoforms generated by alternative promoter usage, where PPARγ1 and PPARγ2 regulate differentially, for instance, the glucose and fatty acid metabolism and the mouse prostate benign epithelial cell differentiation [5].

PPARγ also regulate several metabolic diseases, such as obesity [6], diabetes [7], inflammatory diseases [8,9,10,11] and neuroinflammatory disease [12]. In this review, we focus on PPARγ, highlighting its attributions in pathophysiological processes, mainly those involving lung injury.

## 2. Isoforms and Function of PPARγ

All PPARγ isoforms play an important role in adipocyte differentiation and glucose metabolism; however, their expression is different. The PPARγ1 isoform is expressed in nearly all cells, while PPARγ2 is limited to adipose tissue. Both forms of PPARγ1 and PPARγ2 are essential for the development of adipose tissue and the control of insulin sensitivity. Nevertheless, PPARγ2 is the isoform regulated in response to nutrient intake and obesity [13]. The PPARγ1 isoform is widely expressed in the colon, retina and hematopoietic cells and has also been detected at low levels in other organs, such as the spleen and heart [14].

Two isoforms of PPARγ, γ1 and γ2, differently regulated the mouse prostatic epithelial differentiation. PPAR*γ*1 caused decreased basal cell differentiation and increased tumorigenicity, and PPAR*γ*2 increased basal cell differentiation. [5]. In other studies, it was also demonstrated that PPARγ1 has a role in oncogene and that PPARγ2 acts as a tumor suppressor in prostate cells [15].

## 3. PPARγ Ligands and Overall Effects

PPARγ is activated by several fatty acid metabolites, such as 15-deoxy-D12,14-prostaglandin J2 [16], 9-hydroxyoctadecadienoic acid [17], nitrated fatty acids [18] and lysophosphatidic acid [19]. Other endogenous ligands such eicosapentaenoic acid (EPA) and docosahexaenoic acid (DHA) can bind to PPARγ by their metabolites and modulate the PPARγ expression [20,21].

Compounds from the natural origin are also PPARγ ligands. Researchers identified a small molecular, astragaloside IV, from herbal extracts as a selective PPARγ natural agonist in nervous cells [22]. Another study showed that 6-shogaol attenuates the Gram-negative endotoxin lipopolysaccharide (LPS)-induced inflammation in cells by activating PPARγ [23].

Another group showed that an organosulfur garlic compound, alliin, improved gut inflammation through mitogen-activated kinase/nuclear factor kappa B/activator protein 1/signal transducers and activators of transcription 1 (MAPK-NFκB/AP-1/STAT-1) inactivation and PPARγ activation [24].

Vallée et al. demonstrated the effects of cannabidiol in interactions with the homologous wingless and int-1 (wnt)/β-catenin pathway and PPARγ on oxidative stress and neuroinflammation in Alzheimer’s disease [25]. Studies have also shown that other natural PPARγ agonists can modulate intestinal [26,27] and lung inflammation [28].

Recently, our group demonstrated anti-inflammatory activity in both liver and adipose tissues from septic mice by Mediterranean diet components such as omega 9. Omega 9 is a PPARγ ligand. During sepsis, omega 9 restores PPARγ expression and controls exacerbated inflammation [29,30]. More ligands are presented throughout the text.

As well as certain PPARγ ligands, some natural products behave as PPARγ agonists and antagonists. The natural carotenoid astaxanthin found in seafood acts as a PPAR antagonist, inhibiting PPAR-transactivation activity in lipid-loaded hepatocytes [31].

PPARγ is activated by synthetic agonists, such as the thiazolidinedione (TZD) class of antidiabetic drugs, including pioglitazone and rosiglitazone. TZDs increase insulin sensitivity and improve glycemic control in patients with type 2 diabetes. Nevertheless, they induce adverse effects, such as bone loss, weight gain and fluid retention, which can exacerbate congestive heart failure [32]. In 1999, rosiglitazone was approved for therapeutic use in the United States [33]. Another compound of the thiazolidinediones class, pioglitazone, presented fewer side effects in patients compared to rosiglitazone [34].

Some years later, the increased risk of myocardial infarction and death in patients using those drugs was reported [35].

However, studies showed that pioglitazone attenuated cardiac mobilizations in alloxan-induced diabetic rabbits [36]. According to other studies, researchers have shown that pioglitazone reverses pulmonary hypertension and prevents heart failure via fatty acid oxidation [37].

The elucidation of the PPARγ molecular mechanisms will lead to the selective use in disease therapy [21].

## 4. Mechanisms of Regulation of PPARγ

PPARγ is regulated by several post-translational modifications (PTM), including phosphorylation, small ubiquitin-like modifier (SUMO)ylation, ubiquitination, acetylation and glycosylation. Those PTM are nicely described by Brunmeir and Xu [38]. Most of the studies focus on mechanisms of regulation of PPARγ in cells involved in glucose metabolism. A few reports are discussing the importance of those mechanisms of PPARγ regulation in inflammatory cells. Among these PTM, we can highlight the following.

### 4.1. Phosphorylation

Phosphorylation regulates PPARγ activity, and PPARγ, in turn, phosphorylates, for instance, MAP kinase, and negatively regulates MAP kinase activation.

Studies have shown that the anti-hypertensive drug telmisartan regulated PPARγ phosphorylation and its downstream gene expressions, promoted glucose uptake and acted as an overall insulin sensitizing agent in adipocytes [39]. PPARγ increased expression can be induced by some stimuli, such as peptidoglycan. That phenomenon is regulated by MAP kinase activation [40].

Researchers demonstrated an anti-inflammatory mechanism of galangin in microglia cells stimulated with lipopolysaccharide regulated by PPAR-γ signaling, which will lead to inhibition of the phosphorylation of p38/extracellular signal-regulated kinase (ERK) MAP kinases and consequent inhibition of inflammatory cytokines via the NFκB pathway [41] (Figure 1B). Recently, a study showed an anti-inflammatory activity of a natural product via MAP kinase and PPARγ signaling pathways. *Phelinnus linteus* polysaccharide decreased cytokine expression, regulating PPARγ phosphorylation and MAP kinase pathway inhibition [42]. The plant component flavonoid hyperin is anti-inflammatory. It inhibited cytokine production in LPS-incubated macrophage through a PPARγ-dependent mechanism involving inhibition of the ERK and p38 MAP kinases [43]. The topic “PPARγ interaction with other transcription factor and intracellular signaling proteins” in Section 5 goes further into the role of PPARγ modulating biological effects through phosphorylation.

### 4.2. SUMOylation

Ying and collaborators suggested that SUMOylation of PPARγ by an agonist downregulates chemokine expression through inhibition of NFκB in renal inflammation induced by LPS [44]. The SUMOylation pathway consists in three different proteases, SUMO E1, E2 and E3, that can alter the regulation transcription to target proteins, and the mechanism plays a crucial role in the regulation of cell cycle progression and processes of the tumorigenesis [45] (Figure 1A).

From 2005 onward, there are a few articles about PPARγ SUMOylation and ubiquitination in inflammatory cells. Some of them explore the role of SUMOylation in the anti-inflammatory properties of PPARγ [46,47,48,49].

### 4.3. Ubiquitination

Ubiquitination of PPARγ has only been studied in adipocytes, when the neural precursor cell expressed developmentally downregulated protein 4 (NEDD4), an E3 ubiquitin ligase, interacts with the hinge and ligand binding domains of PPARγ. NEDD4 increases PPARγ stability through the inhibition of its proteasomal degradation [50], and another study showed that the E3 ubiquitin ligase tripartite motif containing 23 (TRIM23) regulates adipocyte differentiation via stabilization of the PPARγ [51] (Figure 1C).

### 4.4. Acetylation

Researchers identified five acetylated lysine residues at positions K98, K107, K218, K268 and K293, of which two (K268ac and K293ac) could be blocked by administration of the TZD rosiglitazone (agonist PPARγ), or by activation of the nicotinamide adenine dinucleotide (NAD_ (-dependent deacetylase sirtuin-1 (SIRT1) deacetylase. In their study, they showed that SIRT1 promotes a beneficial metabolic effect through interaction with PPARγ, leading to insulin sensitization, and implied a therapeutic potential of TZD and SIRT1 agonist combination therapy for obesity [52].

### 4.5. Glycosylation

The β-O-linked N-acetylglucosamine (O-GlcNAc) modification, a post-translational modification on various nuclear and cytoplasmic proteins, is involved in the regulation of protein function. Studies showed that PPARγ is modified by O-GlcNAc in 3T3-L1 adipocytes. In these cells, an increase in O-GlcNAc modification by our inhibitor reduced PPARγ transcriptional activity and terminal adipocyte differentiation. The results suggested that the O-GlcNAc state of PPARγ influences its transcriptional activity, and it is involved in adipocyte differentiation [53].

## 5. PPARγ-Dependent Anti-Inflammatory Mechanisms

Inflammation is the mechanism of human diseases, displaying the five classic inflammatory signs: redness, swelling, heat, pain and subsequent loss of organ function. The inflammation can be distinguishing the following types of inflammation: microbial, autoimmune, allergic, metabolic and physical inflammation, depending on the nature of the irritating cause.

In one inflammatory process, many cells and mediators participate after some inflammatory injury. Some non-immune cells, like skin keratinocytes, mucosal epithelial cells and vascular endothelial cells, act as a first barrier and serve as a sentinel for the exogenous and endogenous causes of inflammation [54]. These cells, together with polymorphonuclear leucocytes (neutrophils, eosinophils, basophils) and the strategically positioned macrophages, dendritic cells, Natural Killer (NK) cells, among others, alert the immune system to the presence of inflammation-causing irritants and modulate the inflammatory response [55,56]. These innate immunity effectors establish a tight communication with B and T cells, constituting adaptive immunity. The effectors provide the signaling relays in inflammation caused by allergies, autoimmune diseases and microbes.

Inflammation as the body’s response to an injury at first would be beneficial, because there would be a mobilization of the innate and adaptive immune system, and this would help to contain the cause of the inflammation, and consequently, the healing of damaged tissues. The “good” side of inflammation [57] will depend on the activity of endogenous suppressors of the inflammatory signaling pathways. Nevertheless, when these physiological suppressors do not work correctly, acute or chronic uncontrolled inflammation can lead to apoptosis, necrosis, fibrosis and ultimately, organ dysfunction at the end of the process [58].

During the process, the adaptative and innate cells respond to proinflammatory injury, producing intracellular and extracellular mediators such as cytokines, chemokines and hematopoietic/vascular growth receptors, while displaying their intrinsic factors. The inflammatory response is perpetuated by self-regulatory feeding loops. Also, intercellular and intracellular inflammatory responses are mediated by cell adhesion molecules, complementary proteins and signal transducers [59,60]. Resolution is also mediated by a wide variety of signals, including cytokines and chemokines, among others [61].

### PPARγ Interaction with Other Transcription Factors and Intracellular Signaling Proteins

The nucleus of the cell is the receptor, processor and propagator of signals that transmit, maintain and extinguish inflammation. At the core, there is a regulatory network that interacts with transcription factors. They regulate genes by binding to their promoters and enhancers, determining the gene profiles of each cell [62,63].

Transcription factors are regulators mobilized to initiate a profound reprogramming of the genome in response to proinflammatory insults. Preventing transcription factors NFκB, AP-1 and STAT1 from going to the nucleus has established their role in inflammation. The pathways have an essential role in the inflammatory process [64].

Many studies show the participation of another nuclear transcription factor, the PPARγ, in inflammatory processes. This factor can negatively modulate other transcription factors in transrepression mechanisms [38].

Studies have already demonstrated that PPARγ has anti-inflammatory effects through innate immune signaling by NFκB, particularly in macrophages. These cells are furthermore capable of producing several PPARγ ligands, which can potentiate the anti-inflammatory pro-resolving actions of this receptor on other cells.

Researchers evaluated the involvement of STAT6 and PPAR-γ signaling during acute schistosomiasis. In this model, the CX3CR-1 chemokine-deficient macrophage enhanced STAT6, leading to the PPARγ signaling to promote macrophages towards M2 polarization, which is an anti-inflammatory and pro-resolutive profile [65].

Another study showed that in an atherosclerosis process, the macrophage molecular signaling and inflammatory responses during ingestion of lipoproteins are modulated by complement protein 1 q (C1q), where this protein suppressed JAK-STAT pathway activation and increased transcriptional activation of PPARγ, coherent as an M2-like polarized response [66].

Geng et al. reported a model of Alzheimer’s disease, showing that the inhibition of miR-128 reduced amyloid-β-mediated cytotoxicity by upregulation of PPARγ and NFκB inactivation in mice neuronal cells and Neuro2a lineage cells [67].

Researchers also showed that suppressing NFκB (p65) protein synthesis and increasing PPARγ gene and protein expression helped magnesium administration to decrease blood glucose levels in diabetic animals [68].

Recently, one group showed that telmisartan, besides its role as an anti-hypertensive drug, has effects against oxidative stress, apoptosis, inflammatory responses and epithelial–mesenchymal transition (ETM). Telmisartan improved oxalate and calcium oxalate crystal-induced EMT by exerting an antioxidant effect through the PPARγ-AKT/STAT3/p38 MAP kinase-Snail signaling pathway. Therefore, telmisartan can block EMT progression and is a potential therapeutic agent for preventing and treating a renal pathology or its recurrence [69].

PPARγ agonist could alleviate intraperitoneal adhesion by regulating macrophage polarization and the suppressor of cytokine signaling proteins (SOCS)/JAK2/STAT/PPARγ signaling pathway [70].

Those reports relating the anti-inflammatory role of PPARγ to the inhibition of other transcription factor activity describe PPARγ’s ability to cause transrepression. The effect of PPARγ on MAP kinases is related to non-genomic activity of PPARγ (Figure 1).

The other anti-inflammatory mechanisms of PPARγ include the prevention of clearance of complexes after SUMOylation, repressing the transcription of inflammatory mediator genes. It involves SUMOylation of the PPARγ–ligand binding domain, keeping the NR co-repressor/histone desacetilase 3 (NCor/HDAC3) complex on the promoter, repressing the transcription of inflammatory genes [46,47]. Instead of acting as a transcriptional activator, SUMOylated PPARγ represses transcription [48,49] (Figure 1).

## 6. Pharmacologic and Therapeutic Potentials of PPARγ Ligands

PPARγ agonists have different effects on a variety of diseases. For instance, the 15d-PGJ2 inhibited tumor progression in vivo [71] and induced apoptosis in tumor cells, suggesting the use of this agonist as an anticancer agent [72]. The PPARγ ligand VSP-77, in a mice model, reduced fasting glucose and insulin levels [73], illustrating its known role as an antidiabetic. Also, PPARγ agonists were linked to reduced chronic obstructive pulmonary disease (COPD) exacerbation rate in diabetic patients, showing PPARγ’s role in lung diseases [74].

Natural compounds may exert their effect through PPAR, as discussed earlier. Curcumin, a spice derived from the rhizome of Curcuma longa Linn [75,76], and magnolol, a natural compound isolated from *Magnolia officinalis* [77,78], showed anti-inflammatory properties.

The compounds targeting PPARγ have their anti-inflammatory activity by inhibiting inflammatory cytokines and activating immune cells. Thus, PPARγ can become a potential therapeutic target for inflammatory bowel diseases, for instance, because PPARγ is highly expressed in the gut [79]. In an animal model, PPARγ reduced hepatic ischemia-reperfusion injury (IRI) and decrease the pro-inflammatory population of Kupffer cells altering macrophage polarization [80]. The family with sequence similarity 3A (*FAM3A)*, a direct target gene of PPARγ, mediates PPARγ protective effects in liver IRI [81]. PPARγ deletion enlarged infarct size, promoted neuron apoptosis and aggravated the ER stress [82]. Likewise, pioglitazone treatment reduced hepatic inflammation and oxidative stress and improved liver function in renal IRI [83]. Thus, PPARγ seems to have a critical role in IRI in the liver, kidney and brain.

A synthetic PPARγ agonist showed a potent anti-inflammatory action modulating cytokine overproduction, proving to be a good candidate for COVID-19 infections [84].

Herein, we further discuss the role of synthetic or natural PPAR agonists in lung inflammatory diseases and bacterial and viral infections, evidencing the potential therapeutic role of those compounds in diverse pathologies involving lung injury.

### 6.1. PPARγ Role in Lung Inflammatory Diseases

IRI can occur with pulmonary thromboembolectomy and lung transplantation, and is characterized by lung inflammation with edema [85]. The use of either rosiglitazone or troglitazone inhibited the IR-induced increase of pro-inflammatory cytokines and neutrophil accumulation in the lung [86].

PPARγ’s role in cancer has excelled. PPARγ hampered tumor development and progression, and controlled the tumor microenvironment, ameliorating tumor growth and metastasis [87]. Treatment with PPARγ agonist and radiotherapy enhanced the effectiveness of tumor control and dampened metastasis [88]. The cyclooxygenase (COX) metabolite prostacyclin acted through PPARγ, promoting anticancer signaling [89]. PPARγ activation transrepressed the NFkB pathway, blocking cell proliferation, differentiation and apoptosis in non-small cell lung carcinoma [90]. PPARγ agonists can be used as monotherapy in lung cancer, or associated with cytotoxic agents [91].

PPARγ has been considered a molecular target for effective asthma therapy [92]. PPARγ negatively regulated the production of mucin and inflammatory mediators by repressing gene expression in primary human bronchial epithelial cells during allergic airway inflammation [93]. Korean red ginseng and *Salvia plebeia* R.Br. alleviated ovalbumin-induced asthma in mice in a fashion dependent on the upregulation of the PPARγ gene and blocking protein kinase B (PKB or Akt) and phosphatase and tensin homolog (PTEN) phosphorylation [94]. The multifaceted anti-inflammatory effects in lung cells during allergic airway diseases [92] point to potential as an adjuvant therapy.

In lung inflammatory disorders, reactive oxygen species (ROS) is a protagonist in diseases such as chronic obstructive pulmonary disease (COPD) and acute respiratory distress syndrome (ARDS) [95]. Cytokines with pro-inflammatory activities are considered biomarkers, predictors of morbidity and mortality during ARDS [96]. LPS induces ROS production and adhesion molecules’ expression. Higher levels of intercellular adhesion molecule-1 (ICAM-1) and vascular cell adhesion molecule 1 (VCAM-1) expression promote the recruitment of leukocytes to the lung, leading to the production of proinflammatory cytokines in the tissue [97]. Curcumin, the dietary polyphenol isolated from the rhizome of turmeric, inhibited NF-κB in COPD, decreasing inflammatory mediator production. Curcumin inhibited NF-κB and upregulated PPARγ activation, ameliorating cigarette smoke extract-induced inflammation in vivo and in vitro [98]. The activation of PPARγ may be an effective therapeutic approach in COPD, as it reduced cigarette smoke-induced inflammation and decreased the magnitude of bacterial infection-caused exacerbations [99].

Cystic fibrosis is an inherited disease with mutations on the cystic fibrosis transmembrane conductance regulator (*CFTR)* gene [100]. A deletion of phenylalanine 508 (F508) affects a high percentage of patients and results in inflammation and other alterations [101,102]. Cystic fibrosis epithelial cells present lower FOXO1 expression [103] and deficiency in PPARγ [104]. Myriocin inhibits ceramide synthesis, reducing inflammation and improving response against infections [105,106]. Transcription factor EB (TFEB) promotes the activation of PPAR and the FOXO family of transcriptional factors involved in lipid homeostasis and inflammatory responses [107]. Treatment with myriocin stimulates nuclear translocation of PPARγ and FOXO1A on F508-CFTR bronchial epithelial cell line IB3-1 cells [108].

LPS-induced endotoxemic shock involves dysregulated inflammation that injures the lungs, and it is often fatal. Endothelial cell PPARγ knockout worsened LPS-induced pulmonary inflammation and injury. There was infiltration of inflammatory cells, edema and ROS and pro-inflammatory cytokine production, with the upregulation of TLR4 expression and activation in lung tissue [109].

HO-1 (heme oxygenase-1), an antioxidant enzyme, is induced by PPAR ligands. PPAR activation and HO-1 can exert therapeutic effects on lung inflammation [110,111]. PPARγ directly regulates HO-1 transcription, impacting inflammation, ROS production and apoptosis [112,113]. The upregulation of HO-1 by PPARγ agonists also inhibits pulmonary cell proliferation and remodeling [114]. HO-1 induction by rosiglitazone via the protein kinase C α (PKC)α/ adenosine monophosphate-activated protein kinase (AMPK)/p38 MAPK/SIRT1/PPARγ pathway suppresses LPS-mediated pulmonary inflammation. Rosiglitazone induces HO-1 expression via either NOX/ROS/c-Src/Pyk2/Akt-dependent Nrf2 activation or PPARγ in human pulmonary alveolar epithelial cells (HPAEpiCs) and suppresses LPS-mediated inflammatory responses, suggesting that PPARγ agonists may be useful for protection against pulmonary inflammation [115]. Upregulation of HO-1 protected against the inflammatory responses triggered by LPS, at least in part, through attenuation of NF-κB [116].

Natural products that bind to PPAR may have a critical role in lung inflammatory response. Wogonin, a flavonoid-like chemical compound found in *Scutellaria baicalensis*, inhibited the production of numerous inflammatory cytokines, including TNFα, IL-1β and IL-6, in the broncoalveolar fluid (BALF) and lung tissues after LPS challenge. The PPARγ inhibitor GW9662 reversed these effects. Wogonin activated PPARγ, which decreased NFκB translocation to the nucleus and binding to DNA in vivo and in vitro [117]. Engeletin (dihydrokaempferol 3-rhamnoside) is a flavanonol glycoside [118]. It can be found in white grapes and white wine. Engeletin activates PPAR-γ and presented protective and therapeutic effects against LPS-induced lung injury [119]. Smiglaside A, a phenylpropanoid glycoside isolated from the traditional Chinese medicinal herb *Smilax riparia*, foster macrophage polarization to an anti-inflammatory M2 phenotype via the AMPK-PPARγ signaling pathway [120]

Endogenous products such as resolvin D1 may also bind to PPAR. Animals who received resolvin D1 stimulated with LPS had lower leukocyte counts and TNF-α and IL-6 levels in BALF compared to the LPS group. Resolvin D1 activated PPARγ and attenuated lung inflammation of LPS-induced acute lung by suppressing NFκB activation [121].

Mesenchymal stem cells are multi-potent non-hematopoietic stem cells residing in most tissues, including the lung [122]. The main beneficial effects reside in the released extracellular vesicles with anti-inflammatory properties. The extracellular vesicles derived from lung mesenchymal stem cells upregulated the PPARγ axis, showing anti-inflammatory and antioxidant effects [123].

Interestingly, recent reports discussed the crosstalk between PPAR pathways with glucocorticoids and adenosine A2A receptor (A2AR). The establishment of ER-Hoxb8-immortalized bone marrow-derived macrophages from Pparg^fl/fl^ and LysM-Cre Pparg^fl/fl^ mice allowed the authors to show the effect of glucocorticoid on PPARγ knockout macrophages. Interestingly, glucocorticoid induces increased recruitment of PPARγ KO, but not PPARγ wildtype macrophages to the site of inflammation. It is a molecular link between glucocorticoids and PPARγ, showing that PPARγ modulates glucocorticoid-induced migration in macrophages [124]. The activation of PPARγ [125,126] and adenosine A2A receptor (A2AR) [127,128] have anti-inflammatory properties in acute lung injury. The A2AR stimulated PPARγ expression via protein kinase A(PKA)– cyclic adenosine monophosphate (cAMP)-response element binding protein (CREB) signaling. PPARγ and A2AR could upregulate the mRNA and protein expressions of each other, generating a positive feedback loop between both increasing their anti-inflammatory effect and reducing lung damages in lung injury [129].

Adipocytokines, such as adiponectin and leptin, are mediators produced mainly by adipocytes and can be regulated by PPARγ [11]. Leptin is a primarily pro-inflammatory adipokine that induced the production of Th1 cytokines (TNF-α, IL-6 and IL-12) [130,131,132] and blocked the production of Th2 cytokines (IL-4, IL-5 and IL-10) [133,134]. The leptin receptor is expressed by human lung cells [135,136,137]. Leptin decreases the expression of PPARγ in rat adipose and liver tissues [138]. Leptin also counteracts PPARγ anti-inflammatory action, which may impact lung inflammatory status during different pulmonary diseases. Adiponectin is predominantly an anti-inflammatory adipokine that inhibits pro-inflammatory cytokines (TNF, IL-6) [139] and induces anti-inflammatory cytokines (IL-10) [140,141]. Adiponectin, adiponectin-activated PPARs and PPAR-induced adiponectin attenuate inflammation [142]. Thus, the balanced release of these adipocytokines driven by PPARγ activation may link adipose tissue to lung pathology [137].

### 6.2. PPARγ’s Role in Bacterial Lung Infection

The alveolar macrophages are key players in pulmonary antimicrobial defense and lung homeostasis [143,144,145,146]. The PPARγ is critical in the alveolar macrophage development and function [147,148,149,150]. The absence of PPARγ in alveolar macrophages boosted Th1-biased inflammation and defective resolution of inflammation [151].

Bacterial infection in the lung induces an inflammatory response that leads to the destruction of the invading pathogen. Nevertheless, the persistence of the inflammation may result in a condition termed non-resolving pneumonia [152]. Bacterial pneumonia is still responsible for a massive number of casualties worldwide [153]. Alveolar macrophages and neutrophils are the major cell types that kill internalized lung bacteria [154]. Mice infected with *Klebsiella pneumoniae* develop lung injury with an accumulation of cardiolipin, the main lipid of the inner mitochondrial membrane. High concentrations of cardiolipin have been detected in the lung fluid of patients with pneumonia [155]. Cardiolipin induces SUMOylation of PPARγ at K107, which is distinct from the SUMOylation rosiglitazone [47]. Cardiolipin-induced SUMOylation inhibits IL-10 production by lung CD11b þLy6GintLy6CloF4/80 þ cells because of the recruitment of a repressive nuclear receptor corepressor (NCOR)/ histone deacetylase 3 (HDAC3) complex to the IL-10 promoter, ending in persistent inflammation during pneumonia [156].

The pathogenic *P. aeruginosa* has diverse immune system evading mechanisms, including the production of virulence factors and biofilm formation. PPARγ agonists play a pivotal role in the host response to virulent *P. aeruginosa* [157]. Quorum sensing is a mechanism wherein the bacteria secrete small molecules, such as N-(3-oxo-dodecanoyl)-l-homoserine lactone (3O-C12-HSL), that promote biofilm formation and interbacterial communication [158]. Quorum sensing genes in *P. aeruginosa* (strain PAO1) and 3O-C12-HSL attenuate PPARγ expression in bronchial epithelial cells, with loss of barrier integrity and decreased junctional proteins (occludin, claudin-4), which is restored by PPARγ agonists [157]. So, the use of PPARγ agonists can serve as an adjuvant in treating resistant *P. aeruginosa* infections.

The absence of PPARγ in lung macrophages reduces the growth of virulent *Mycobacterium tuberculosis*, enhances proinflammatory cytokines and reduces granulomatous infiltration—PPARγ activation—which downregulates macrophage proinflammatory responses and enables *Mycobacterium tuberculosis* growth [159]. A murine model of multiwall carbon nanotube (MWCNT) elicited chronic granulomatous disease, which resembles human sarcoidosis pathology, including alveolar macrophage PPARγ deficiency. PPARγ deficiency promotes pulmonary mycobacterial early secreted antigenic target protein 6 (ESAT-6) retention, exacerbates macrophage responses to MWCNT + ESAT-6 and intensifies pulmonary fibrosis [160]. The cells overexpressing CYP1A1 infected with *Micoplasma hyopneumoniae* led to an increase in PPARγ expression, resulting in lower production of IL-1b, IL-6, IL-8 and TNF-α in pigs [116].

Sepsis is a systemic inflammatory syndrome in response to an infection [161]. Pioglitazone decreased inflammatory response in polymicrobial sepsis targeting the NFκB pathway, which reduced pro-inflammatory cytokines levels [162]. Sepsis causes changes in PPARγ expression and activation [29], in part because of phosphorylation of PPARγ by ERK1/2. ERK1/2 inhibition reversed PPARγ phosphorylation, ameliorating lung injury [163].

Triggering receptor expressed on myeloid cells 2 (TREM)-2 macrophages [164] negatively regulate toll-like receptor (TLR)-mediated responses and enhance phagocytosis. TREM-2 enhances bacterial elimination and improves survival in a sepsis model [165]. Trem-2^-/-^ mice infected with *S. pneumoniae* exhibited an augmented bacterial clearance from the lungs. The increased levels of C1q were crucial for enhanced bacteria phagocytosis in a mechanism dependent on PPARγ activity [166].

### 6.3. PPARγ Role in Viral Lung Infection

The PPARγ agonists TZDs have ameliorating effects on severe viral pneumonia. Also, diet ligands of PPARγ, like curcuma, lemongrass and pomegranate, have anti-inflammatory properties through PPARγ activation, acting as immunomodulators regulating cytokine levels [84]. In this regard, both natural and synthetic PPAR-γ ligands treatment declined host morbidity and mortality during influenza A virus infection [78,167,168,169,170]. PPARγ expression was downregulated in resident macrophages during influenza A virus infection. A phytohormone abscisic acid binds to the G-protein coupled receptor lanthionine synthetase component C-like 2, increasing cAMP levels [171]. The abscisic acid modulates immune and inflammatory responses in mouse models of colitis and obesity [172]. Also, abscisic acid improves influenza virus-induced pathology by activating PPARγ in lung immune cells [173]. The PPARγ activation by the 15d-PGJ2 protects mice against influenza virus infection with reduced lung cytokine production [169]. The myeloid PPARγ expression in alveolar macrophages is critical for modulating pulmonary inflammation, the development of host diseases and the recovery of tissue homeostasis following respiratory viral infections [174].

The genetic-induced obese (db/db) mice were more susceptible to viral infection with higher viral replication, higher inflammatory response and damaged lung repair after influenza infection, with an increased mortality rate. PPARγ was downregulated in the lung macrophages of db/db mice after influenza infection. Strikingly, the treatment with 15d-PGJ2 protected the db/db mice after influenza infection [175]. Thus, treatments with PPARγ agonists may be a potential candidate to treat influenza infection in obese patients.

The coronavirus disease 2019 (COVID-19) and type 2 diabetes are two pandemic diseases with enormous health and economic costs. Some COVID-19 patients may evolve to severe pneumonia with a high fatality rate, especially in patients with chronic conditions, such as diabetes and cardiovascular disease. Severe acute respiratory syndrome coronavirus (SARS-CoV-2) binds to the angiotensin-converting enzyme 2 (ACE2). In an obesity animal model, pioglitazone increased the expression of ACE2 in liver, adipose tissue and skeletal muscle [176]. Rosiglitazone also induced an increase in ACE2 expression [177]. Those drugs are broadly adopted in type-2 diabetes [178]. Therefore, the use of this PPARγ agonist may be used with caution during the COVID-19 pandemic.

Using data from the National Health and Nutrition Examination Survey (NHANES) between 1988–1994 and 1999–2010, it was shown that influenza/pneumonia mortality was associated with the medication. Patients taking rosiglitazone had an increased risk of mortality from influenza/pneumonia. Although most influenza infections do not represent a health burden, secondary bacterial infection results in high rates of mortality and morbidity. Viral infection, such as influenza, predisposes the lung to secondary bacterial infection by dysregulation of the host immune response [179,180,181,182]. Community-acquired methicillin-resistant *Staphylococcus aureus* (MRSA) represents a severe life threat during influenza-associated secondary bacterial infection [183,184]. Rosiglitazone has an anti-inflammatory effect during acute pulmonary inflammation [185]. Rosiglitazone treatment compromised bacterial clearance during influenza-bacterial super-infection, thus worsening the outcome of influenza-associated pneumonia [186]. The use of natural or synthetic anti-inflammatory compounds must be used with caution during viral infection so as to prevent potential future patients’ immune response derangement and increased risk of infections.

A simplified scheme of PPARγ actions in the lung is shown in Figure 2.

After a lung injury, an inflammatory process is observed, with an increase in proinflammatory cytokines and edema formation. After binding to its agonist, PPARγ inhibits transcriptional factor, leading to a decrease in proinflammatory mediators and to early resolution of the inflammation.

## 7. Concluding Remarks Considering Experimental Findings and Clinical Trials

Most of the reports discuss the anti-inflammatory role of PPARγ agonists on cytokine production induced by many agents that cause lung inflammation or infection. Those effects are related to the PPARγ-induced transrepression mechanisms by inhibiting transcription factors or by crosstalk with other receptors or adipocytokines with a beneficial effect on different diseases. This evidence highlights PPARγ as a potential target for adjunct therapy in lung diseases.

Clinical studies with PPARγ agonists based on experimental findings show their effects on disease outcome. Experimental evidence looks promising because of the broad effects of PPARγ ligands controlling the function of many cell types, regulating metabolism and immune response. PPARγ induces and represses transcription, interacts with co-activators and co-receptors and has non-genomic effects. The broad gamma of mechanisms and multiple targets reinforces PPARγ’s role in controlling many biological effects. PPARγ binds to structurally different ligands at different sites of its binding pocket, a phenomenon that may explain various effects, and is also used as a target for new drug design studies.

The role of pioglitazone in metabolic syndrome is well established, because it reduced the risk of cardiovascular events in diabetic insulin-resistant patients, and decreased the progression of atheroma and even diabetes incidence [187].

PPARγ agonists induce differentiation and apoptosis, and prevent proliferation, effects that may help to decrease cancer incidence and progression [87,91]. Pioglitazone associated with imatinibe diminished cancer development [188]. Clinical evidence shows synergism between PPARγ agonist with a chemotherapeutic agent in lung cancer [91].

Some clinical trials show that the use of PPARγ agonists should be analyzed with caution. In a randomized, placebo-controlled, double-blinded, crossover clinical trial, pioglitazone did not improve the primary outcome score in severe asthmatics [189]. There was no efficacy of pioglitazone treatment for 12 weeks in mild asthma management [190].

At the site of the binding pocket, the PPARγ ligand binding affects the PPARγ effect, and we may also consider the dose and the duration of the treatment. Studies with long-duration treatment were not conclusive and showed no benefits, but acute administration of PPARγ agonists was associated with a reduction in the risk of pneumonia, suggesting a beneficial effect of acute treatment with PPARγ agonists [11]. Concerning bacterial infections, as in cancer, the best result is obtained with the use of the PPARγ agonist as an adjuvant treatment, associated with the current treatment, as antibiotics in the former case [11,191]. By blocking immune response, PPARγ agonists’ chronic prescription must be used with caution to prevent bacterial and viral infections.

More than 500 trials with pioglitazone, considered the safest PPARγ agonist in terms of side effects, have been performed worldwide and are still under development [192]. More studies and clinical trials are needed in the field.

## Figures and Tables

**Figure 1 ijms-22-00805-f001:**
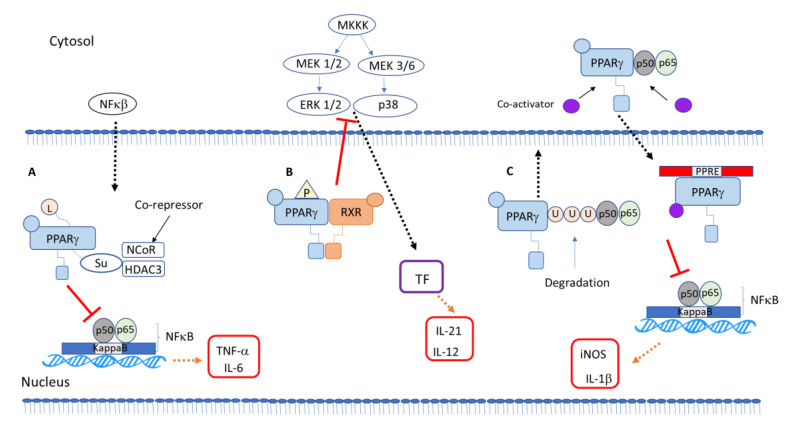
Molecular regulatory mechanisms of PPARγ. (**A**) A ligand-dependent transrepression. Binding to PPARγ ligand leads to SUMOylation of PPARγ, which stabilizes the co-repressor, leading to blockade of NFκB target gene expression. (**B**) Non-genomic role of PPARγ. PPARγ phosphorylates MAP kinase, leading to inhibition of transcription factor (TF) binding. (**C**) Transrepression. ubiquitination of PPARγ increases its stability, and it can also inhibit NFκB target genes. PPARγ (peroxisome proliferator-activated receptor γ), small ubiquitin-like modifier (SUMO)ylation, NFκB (nuclear factor kappa B).

**Figure 2 ijms-22-00805-f002:**
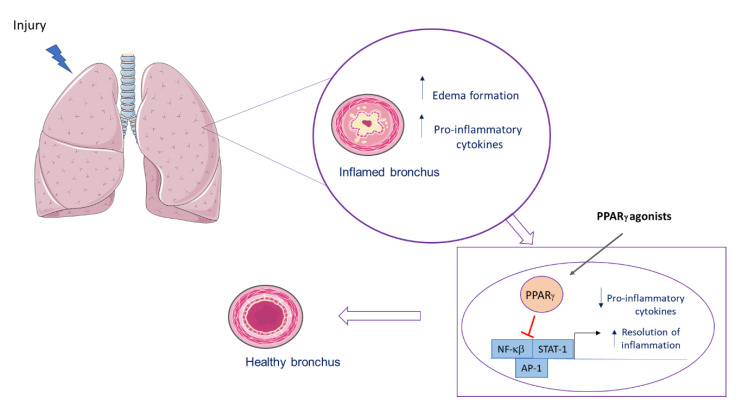
PPARγ role in pulmonary inflammation.

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
