# Peer review of "PPAR Gamma: From Definition to Molecular Targets and Therapy of Lung Diseases"

_ijms, 2021, doi:10.3390/ijms22020805_

Round 1

Reviewer 1 Report

The review article " PPAR gamma: from definition to molecular targets and therapy of lung diseases" tries to give an overview of the role of PPAR gamma in pathogenesis and treatment of patients with lung diseases. The topic of this review is interesting. However, in my opinion the conclusions are not clearly enough defined. The authors should discuss what are the key findings in the research done in this field so far.

Author Response

We are glad to have the opportunity to respond to the reviewer comments of our manuscript entitled: PPAR gamma: from definition to molecular targets and therapy of lung diseases, by Márcia Vidal de Carvalho and Cassiano Felippe Gonçalves de Albuquerque and Adriana Ribeiro Silva. We hope we have covered all suggestions in the revised version of our manuscript.

Reviewer 1:

The review article " PPAR gamma: from definition to molecular targets and therapy of lung diseases" tries to give an overview of the role of PPAR gamma in pathogenesis and treatment of patients with lung diseases. The topic of this review is interesting. However, in my opinion the conclusions are not clearly enough defined. The authors should discuss what are the key findings in the research done in this field so far.  

Answer: We appreciate the reviewer finds the topic of our review article interesting. The conclusions were not clear. We believe we have improved and clarified the conclusions adding a discussion of our view of the key findings in the field in the concluding remarks topic.

Reviewer 2 Report

This is a well-stuctured and concise review on the significant role of PPΑRγ in lung pathology. Its molecular targets and mechanisms of action are adequately described.

Minor comments to the authors:

1)PPARγ is also involved in other inflammatory lung diseases such as ischemia-reperfusion injury, allergic airway inflammation, and cancer. A small section revising these (under the heading 6.1) might be also included by the authors.

2) The therapeutic efficacy of PPARγ pharmacological ligands in selected pulmonary vs nonpulmonary diseases may also be further clarified/discussed

3)Certain PPARγ ligands exert agonistic and antagonistic activity towards the PPARγ receptor. Whether natural products targeting PPAR behave similarly needs to be discussed.

4) Activated PPARγ in adipocytes ensures a balanced secretion of adipocytokines (adiponectin and leptin) that are mediators of actions in peripheral tissues. An up-to date recap on the mechanistic links between adipocytokines and lung pathology (Ali Assad N, Sood A. Leptin, adiponectin and pulmonary diseases. Biochimie. 2012;94(10):2180-2189.) should be also covered in a small paragraph

5) The abstract and the manuscript need to be revised by a native English speaker for minor syntaxis/grammatical errors. 

Author Response

We are glad to have the opportunity to respond to the reviewer comments of our manuscript entitled: "PPAR gamma: from definition to molecular targets and therapy of lung diseases", by Márcia Vidal de Carvalho and Cassiano Felippe Gonçalves de Albuquerque and Adriana Ribeiro Silva. We hope we have covered all suggestions in the revised version of our manuscript.

Reviewer 2:  

This is a well-structured and concise review on the significant role of PPΑRγ in lung pathology. Its molecular targets and mechanisms of action are adequately described.

Answer: We tried our best to structure and to make the review concise. One of our main goals was to describe the molecular targets and mechanisms of action adequately. Thanks for mentioning it.

Minor comments to the authors:

1) PPAR
γ is also involved in other inflammatory lung diseases such as ischemia-reperfusion injury, allergic airway inflammation, and cancer. A small section revising these (under the heading 6.1) might be also included by the authors.

Answer: Thanks for pointing that out. It was missing, and we have not realized that before. We included a small section under the heading 6.1 revising PPARγ involvement in other inflammatory lung diseases such as ischemia-reperfusion injury, allergic airway inflammation, and cancer.

2) The therapeutic efficacy of PPARγ pharmacological ligands in selected pulmonary vs nonpulmonary diseases may also be further clarified/discussed

Answer:  It is very good subject. We discussed the efficacy of PPARγ ligand therapeutic use in selected pulmonary vs nonpulmonary diseases under the heading 7, and we believe that this suggestion has added great value to the manuscript.

3) Certain PPARγ ligands exert agonistic and antagonistic activity towards the PPARγ receptor. Whether natural products targeting PPAR behave similarly needs to be discussed.

Answer: We added the discussion implying that as certain PPARγ ligands, some natural products behave as PPARγ agonists and antagonists under the heading 3.

4) Activated PPARγ in adipocytes ensures a balanced secretion of adipocytokines (adiponectin and leptin) that are mediators of actions in peripheral tissues. An up-to date recap on the mechanistic links between adipocytokines and lung pathology (Ali Assad N, Sood A. Leptin, adiponectin and pulmonary diseases. Biochimie. 2012;94(10):2180-2189.) should be also covered in a small paragraph.

Answer: We mentioned and discussed that matter in a previous publication. Thanks for the opportunity to discuss and contextualize it in the manuscript. The balanced secretion of adipocytokines depends on activated PPARγ. Those are mediators with key actions in peripheral tissues. The mechanisms involved in lung pathology-related to adipocytokine effects needed to be discussed. We included a small paragraph under heading 6.1.

5) The abstract and the manuscript need to be revised by a native English speaker for minor syntaxis/grammatical errors. 

Answer: We have carefully revised the manuscript for minor syntaxis/grammatical English errors.